# Parameter Screening of PVDF/PVP Multi-Channel Capillary Membranes

**DOI:** 10.3390/polym11030463

**Published:** 2019-03-11

**Authors:** Jan O. Back, Rupert Brandstätter, Martin Spruck, Marc Koch, Simon Penner, Marco Rupprich

**Affiliations:** 1Department of Environmental, Process & Energy Engineering, MCI—The Entrepreneurial School, Maximilianstrasse 2, 6020 Innsbruck, Austria; jan.back@mci.edu (J.O.B.); rupertbrandstaetter@yahoo.com (R.B.); martin.spruck@mci.edu (M.S.); marc.koch@mci.edu (M.K.); 2Institute of Physical Chemistry, University of Innsbruck, Innrain 52c, 6020 Innsbruck, Austria; simon.penner@uibk.ac.at

**Keywords:** multi-channel membrane, PVDF/PVP membrane, phase inversion process, design of experiments

## Abstract

The increasing research in the field of polymeric multi-channel membranes has shown that their mechanical stability is beneficial for a wide range of applications. The more complex interplay of formation process parameters compared to a single-channel geometry makes an investigation using Design of Experiments (DoE) appealing. In this study, seven-channel capillary membranes were fabricated in a steam–dry–wet spinning process, while varying the composition of the polymer solution and the process temperatures in a three-level fractional factorial linear screening design. The polymers polyvinylidene flouride (PVDF) was the chemically resistant main polymer and polyvinylpyrrolidone (PVP) was added as hydrophilic co-polymer. Scanning electron microscopy and atomic force microscopy were applied to study the membrane morphology. Fabrication process conditions were established to yield PVDF/PVP multi-channel membranes, which reached from high flux (permeability *P* = 321.4
L/$m^{2}$/h/bar, dextran 500 kDa retention *R* = 18.3%) to high retention (*P* = 66.8
L/$m^{2}$/h/bar, *R* = 80.0%). The concentration of the main polymer PVDF and the molecular weight of the co-polymer PVP showed linear relations with both *P* and *R*. The permeability could be increased using sodium hypochlorite post-treatment, although retention was slightly compromised. The obtained membranes may be suitable for micro- or ultra-filtration and, at the same time, demonstrate the merits and limitations of DoE for multi-channel membrane screening.

## 1. Introduction

Membrane technology has received increasing attention over the past decades due to its many advantages in various industrial processes. Frequently mentioned benefits are relatively low energy costs, no required phase transition or chemical addition, easy scale-up, and simple operation due to the modular design. As a result, membrane technology is a key element in the fields of water treatment, food and pharmaceutical processing, and gas purification [1]. The main market share is held by polymeric membranes due to their versatility and inexpensive production [2], whilst the polymer polyvinylidene flouride (PVDF) is one of the most promising materials due to its robust mechanical strength, favorable thermal stability, and—most importantly—excellent chemical resistance [3,4,5].

In an attempt to create a sufficiently large filtration area, membrane modules have been developed, which feature either a tubular or flat sheet membrane arrangement. Tubular membranes, in particular capillaries or hollow fibers, offer several merits over the flat sheet configuration, such as higher productivity per module volume, self-supporting structures, and easy handling in module production through, for instance, the redundancy of spacers [2,6,7]. While hollow fibers (diameter 0.05 mm to 0.5 mm) are mainly driven outside-in, capillary membranes (diameter 0.25 mm to 2 mm) can be operated inside-out with reduced risk of fiber blockage through particle load, which creates high shear forces at the channel surfaces and mitigates membrane fouling [8].

Multi-channel membranes with up to 60 feed channels were developed from ceramics in the late 1970s in order to further increase the filtration area to volume ratio [9,10]. They were later refined [11,12,13] and provide better tensile strength in both radial and axial direction compared to their single-channel counterparts [14]. Polymeric multi-channel membranes started to emerge from polymer melt extrusion, which was first patented by Toray to obtain hollow tape-shaped membranes [15], and afterwards further investigated to realize non-porous micro-capillary films from polyethylene [16] to be applied for heat exchange [17,18].

Polymeric non-solvent induced phase separation (NIPS) membranes with multi-channel geometry and ultrafiltration (UF) characteristics have been commercialized by Inge (now BASF) [19] and GE Water (now SUEZ Water Technology and Solutions) [20], who both use polyethersulfone (PES) and seven feed channels, and by Hyflux [21], applying PES or PVDF as main polymer with three feed bores. Inge membranes [19] have been employed for treatment before seawater desalination [22,23,24,25], and research has also been conducted in channel geometry optimization [26], operational aspects [27], and anti-fouling modification [28]. The pre-treatment for reverse osmosis (RO) has also been proven for Hyflux membranes [21] from PVDF [29].

In the past few years extensive research on multi-channel phase inversion membranes has been conducted by research groups around Chung. Firstly, rectangular seven channel films from PVDF and clay additive were developed for membrane distillation (MD) [30]. Polyacrylonitrile (PAN) membranes for UF were produced using the same geometry [31]. A seven-bore concentric structure spun from PVDF and the additives ethylene glycol (EG), polyethylene glycol (PEG), and LiCl was presented for MD [32,33], showing that the burst pressure improves 67% compared to single-bore fibers [34]. A tri-bore geometry has been proposed for UF and forward osmosis (FO) membranes from Matrimid^®^ [35,36] and for oil/water UF from sulfonated polyphenylenesulfone [37]. More research on the tri-bore design was carried out with the material P84 (co-polyimide) and a polyamide layer (PA)/polyglycerol grafting for FO/oily water separation, respectively [38]. A membrane from a blend of PVDF, polytetrafluoroethylene (PTFE), and EG was dip-coated with Teflon^®^ AF 2400 for MD [39]. Isopropanol dehydration was achieved through pervaporation membranes from Ultem^®^ (polyetherimide) and a PA layer [40].

Besides, research in the field of multi-channel membranes produced rectangular 19-bore ethylene-co-vinyl alcohol membranes with polyvinylpyrrolidone (PVP) additive for potential membrane bioreactor (MBR)/microfiltration (MF)/UF application [41]. Another tri-bore membrane from PVDF was published for air dehumidification [42]. A five-bore membrane design was proposed from blends of PVDF, polymethyl methacrylate (PMMA) and thermoplastic polyurethane (TPU) [43], and from PVDF/PMMA/PVP, without reporting the PVP molecular weight [44,45]. Most recently, the blend PVDF/PEG with *N*-Methyl-2-pyrrolidone (NMP) as solvent was used to spin seven-bore membranes for UF [46], and tri-bore membranes for the control of dissolved oxygen in aquaculture water [47]. In previous studies we reported seven-channel membranes from PES/PVP blends and a PA coating on the lumen side for nanofiltration (NF), refining the spinning process, spinneret design, and coating procedure [48,49].

In this study, we aim to produce and characterize seven-channel capillary membranes for MF/UF in a NIPS spinning process from the system PVDF—main polymer with high chemical resistance/PVP—hydrophilic and pore-forming additive [50]/*N*,*N*-dimethylacetamide (DMAc)—solvent. As numerous factors control and influence the spinning process and final membrane characteristics (concentration of PVDF in spinning dope *c*_PVDF_ [50,51], concentration of PVP in spinning dope *c*_PVP_ [50,52,53,54], temperature of spinning dope *T*_Dope_ [55,56], temperature of coagulation bath *T*_H_2_O_ [55,56], PVDF molecular weight $M_{W}$_PVDF_ [57,58], and PVP molecular weight $M_{W}$
_PVP_ [50]), we used a fractional factorial design generated via Design of Experiments (DoE) to reduce the number of spinning assays. The chosen screening design allows the study of linear relations, but is not suitable for optimization purposes. The effectiveness of DoE in membrane research has been demonstrated for MD with hollow fibers [59,60], but has not previously been applied to multi-channel membranes. However, it is agreed that the formation mechanism of multi-channel membranes is far more complex compared to single-channel membranes, which makes the usage of DoE attractive in this research field. Moreover, we included individual experiments to study the influence of a varied bore-fluid composition and post-treatment via sodium hypoclorite.

## 2. Materials and Methods

### 2.1. Chemicals and Instruments

Multi-channel capillary membranes (MCM) were prepared from spinning dope solutions containing polyvinyledene flouride (PVDF) Solef 6012, 1015, and 6020 with mean weight average molecular weights ($M_{W}$) of 390 kDa, 585 kDa, and 685 kDa, respectively, from Solvay (Schulmann GmbH, Kerpen, Germany) as the main polymer. The co-polymer polyvinylpyrrolidone (PVP) with a $M_{W}$ of 10 kDa, 55 kDa (both from Sigma-Aldrich, Vienna, Austria), and 360 kDa (PVP-K90, Carl Roth, Karlsruhe, Germany), respectively, was added and *N,N*-dimethylacetamide (DMAc, >99%) from Sigma-Aldrich was used as solvent to obtain PVDF/PVP/DMAc spinning dopes. The bore-fluid composition was varied via ethanol (EtOH, >96%, Carl Roth) and DMAc addition to tap water. A sodium hypochlorite (Carl Roth) solution with 12% free chlorine diluted in deionized water to 5000 ppm NaClO was used as a post-treatment solution.

Dextran with a $M_{W}$ of 500 kDa (Carl Roth) was used for macromolecule retention *R* experiments. The feed concentration was adjusted to 200 ppm and the relative drop in carbon content of the solution was measured at 1 bar transmembrane pressure (TMP) with a total organic carbon analyzer (TOC-VCPN) from Shimadzu (Korneuburg, Austria). The permeability *P* was measured as pure tap water permeability at 1 bar TMP. For both *R* and *P* experiments a constant feed temperature of (25±1)
${}^{\circ }C$ was provided via a heating/cooling jacket.

The cross-sectional morphology of the fabricated MCM was studied by scanning electron microscopy (SEM) (JEOL, NeoScope JCM-5000, Freising, Germany). Fibres were cryofractured in liquid nitrogen to obtain a smooth observation surface in radial direction and the samples were studied without previous electro-conductive coating. Atomic force microscopy (AFM) of the inner channel surface was performed using a Veeco Instruments (Mannheim, Germany) Dimension 3100 system with a Nanoscope IVa (Veeco Instruments) controller to study the topography of the prepared MCM. The AFM was operated in tapping-mode at room temperature in air, and microfabricated probes (Tap190-G, Budgetsensors, Sofia, Bulgaria) of monolith silicon with cantilever resonance frequencies around 190 kHz and a spring constant around 48 N/m were used. The rotated tip with a polygon-based pyramidal shape had a height of approximately 17 μm, a setback from the free end of the cantilever of 15 μm and a radius of curvature <10 nm. For surface topography plots and mean roughness calculations, the data was leveled by mean plane subtraction and third-order polynomial background removal, using the software Gwyddion (v2.51) [61].

A rotational viscometer (Rheolab QC, measurement system CC17, Anton Paar, Graz, Austria) was used to measure the rheological properties of the PVDF/PVP/DMAc dope solutions. The applied shear rate $\dot{\gamma }$ was varied in the range of 10 s^−1^ to 1000 s^−1^, while the temperature was kept constant at the respective temperature of the spinning dope *T*_Dope_ given by the screening design (Table 1).

### 2.2. Preparation of PVDF/PVP Multi-Channel Membranes

The degassed PVDF/PVP/DMAc dope solutions were subsequently used for MCM preparation in a steam–dry–wet spinning process. The dope solution, kept at different temperatures *T*_Dope_, was introduced into a specially designed spinning nozzle, which was also heated to *T*_Dope_, with axisymmetric flow and minimum deflection of the dope to avoid uncontrollable shear stress and to ensure molecular orientation. The spinneret consisted of seven concentric orifices (one in the center, six arranged radially around the center) for the bore-fluid, which acted as internal coagulant at a temperature of 30 ${}^{\circ }C$. The outer diameter of the dope outlet was 4.8
mm and the outer diameter of the bore-fluid orifices was 0.9
mm. The nascent fibres passed a steam and air gap of 15 cm before dropping into the external coagulation bath (tap water at different temperatures *T*_H_2_O_). A wind-up unit in the coagulation bath provided continuous MCM production, while not drawing the fibres in the steam/air gap, i.e., a wind-up speed of approximately 11 cm/s. A more detailed description of the applied spinning process is described in [49].

The resulting PVDF/PVP MCM were stored for 5 min in the coagulation bath before being transferred into the post-treatment agent, in which they were stored for 1 d. As post-treatment solution tap water was used for the experiments generated via DoE. Furthermore, we individually studied the effects of sodium hypochlorite post-treatment and addition of organic compounds (EtOH and DMAc) to the bore-fluid. For the post-treatment study, three randomly selected batches, in which the $M_{W}$
_PVP_ was 10 kDa, 55 kDa, or 360 kDa, were stored also in a 5000 ppm NaClO solution. For the variation of the bore-fluid composition, solutions of EtOH 1:1 H_2_O and DMAc 1:1 H_2_O were used as inner coagulant and tap water was used as post-treatment agent. After post-treatment in either water or hypochlorite solution, the fibres were rinsed and flushed several times with water and processed in a wet state to modules containing a single multi-channel membrane with an approximate length of 54 cm, of which roughly 14 cm were used for the potting procedure, resulting in 30 cm active filtration length.

### 2.3. Parameter Screening Design

As numerous factors affecting the membrane properties of the applied polymeric system PVDF/PVP have been studied in literature [3,5,6] for simpler geometries (i.e., flat/hollow fibre membranes), the effect of each parameter on MCM characteristics was not studied individually here; instead, DoE was used, drastically reducing the number of required experiments. A three-level fractional factorial linear screening design with three centre points was generated with the software package MODDE (v12.1, Sartorius Stedim Data Analytics AB, Umea, Sweden [62]). The range of the process variables was derived from literature [50,51,52,53,54,55,56,57,58], delimiting the design space with 4 variables set to quantitative (concentration of PVDF in spinning dope *c*_PVDF_ 12–18 wt.%, concentration of PVP in spinning dope *c*_PVP_ 4–6 wt.%, temperature of spinning dope *T*_Dope_ 20 ${}^{\circ }C$ to 50 ${}^{\circ }C$, and temperature of coagulation bath *T*_H_2_O_ 20 ${}^{\circ }C$ to 50 ${}^{\circ }C$) and 2 factors set to multilevel ($M_{W}$
_PVDF_ 390, 585, 685 kDa and $M_{W}$
_PVP_ 10, 55, 360 kDa). The generated screening design with 21 possible dope solutions is tabulated in Table 1. The models were fitted via multiple linear regression.

## 3. Results and Discussion

### 3.1. Parameter Screening

#### 3.1.1. Dynamic Viscosity

The dope solution’s rheological properties are of major importance to the kinetics and thermodynamics of the phase inversion process and, hence, to the final membrane characteristics [51,53]. The basic statistics of the viscosity measured at $\dot{\gamma }$ = 100 s^−1^ are displayed in Table 2. The dope solutions generated by DoE in Table 1 comprised 21 experiments, but five runs had to be excluded from subsequent spinning experiments, as in one case no homogeneous solution was formed, and in the other cases, the viscosity was too low for spinning. A very low viscosity, under approximately 2000 mPa
s, leads to instabilities in the presented spinning process.

It is noted that all polymeric dope solutions showed shear-thinning behavior, which is in accordance with other studies [51]. Even though the shear and elongation viscosities during MCM spinning may deviate from the single-point measurement at $\dot{\gamma }$ = 100 s^−1^ of the apparent viscosity, this value is thought to be a representative estimate for the solution’s rheological properties.

The model for the dynamic viscosity at $\dot{\gamma }$ = 100 s^−1^ fitted via multiple linear regression is presented in Table 3. The response, i.e., the dynamic viscosity, is calculated from the studied factors and computed coefficients as follows [62]:(1)$$\eta (\dot{\gamma }\;=\;100\;s^{-1})=\mathrm{Constant}+c_{\mathrm{PVDF}}\;\cdot \;{\mathrm{Coeff}}_{1}+c_{\mathrm{PVP}}\;\cdot \;{\mathrm{Coeff}}_{2}+T_{\mathrm{Dope}}\;\cdot \;{\mathrm{Coeff}}_{3}+\cdots$$

As in the measurement of the dope solution’s viscosity the temperature of the coagulation bath *T*_H_2_O_ does not apply, it is not fitted in this model. All other factors were found to have a statistically significant linear influence on the dynamic viscosity, and also the regression model itself is significant. Furthermore, the summary of fit in four parameters from MODDE is listed—ranging from 0 to 1, whereas 1 is perfect. Relatively high model fit $R^{2}$ and future prediction precision $Q^{2}$—both close in size—indicate a good model. Model validity is low—a value less than 0.25 indicates various statistically significant model problems—but reproducibility is high. The latter indicates a low variation of the three replicates of the centre point (3211, 3326, and 3669 mPa
s) compared to the overall variability. This may result in low model validity [62].

Taking into account this information, the presented model describing the dynamic viscosity is rated as reasonable, even though only a screening design with a reduced number of experiments has been used. Consequently, the model allows for the prediction of the viscosity of dope solutions with varied composition and other parameters within the confined design space. However, it should be pointed out that the model is strictly linear, i.e., higher order dependencies could not be tested for, and that the question of mutual dependencies could not be accounted for in the chosen design. Non-linearities, however, do occur when the polymer content is varied over wider ranges than observed here [51].

Furthermore, the coefficients of linear regression model for the viscosity seem sensible insofar that they may be explained via physical models. Higher polymer concentrations lead to a greater degree of chain entanglement [51], a higher temperature increases the chance of surpassing the activation energy of polymer chain attraction through London dispersion and other interactions, and the molecular weight—i.e., the chain length—also leads to stronger chain entanglement.

#### 3.1.2. Permeability and Macromolecule Retention

PVDF/PVP multi-channel membranes were prepared via steam–dry–wet spinning while varying the process temperatures and the composition of the spinning dope solution. The multi-channel membrane preparation succeeded for all of the 16 tested spinning solutions insofar that the spinning process could be carried out successfully and all seven feed channels developed uniformly and circularly, as shown in later Section 3.1.3. This is mostly ascribed to the advanced spinning nozzle design, which avoids uncontrolled shear stress and provides uniform molecular orientation [63].

Membrane permeability and macromolecule retention are decisive for its performance and field of application. Table 2 shows descriptive statistics of both the pure water permeability—calculated per surface area of the seven feed channels—and retention of dextran ($M_{W}$ = 500 kDa). The membrane characteristics range from high flux (*P* = 321.4
L/m/h/bar and *R* = 18.3%) to high retention (*P* = 66.8
L/m/h/bar and *R* = 80.0%). The figures indicate that the produced PVDF/PVP MCM are on the borderline between macro- and ultrafiltration [8].

The multiple linear regression models for *P* and *R* are tabulated in Table 3. Both models show statistical significance, but the only significant factors are the concentration of the main polymer, *c*_PVDF_, and the molecular weight of the co-polymer, $M_{W}$
_PVP_. Hence, these factors display a linear relation with *P* and *R* within the studied range. This is reasonable, as a higher PVDF content leads to a lower porosity and, thus, lower permeability and higher retention [50,52,53,54]. Low molecular weight PVP is leached out from the membrane matrix during the phase inversion process and leaves behind an open-porous network, i.e., high permeability and low retention. High molecular weight PVP partially remains entrapped in the membrane matrix, obstructing the pore interconnection path and leading to lower permeability [50]. Unlike Wang et al. [50], who found that high molecular weight PVP also led to lower retention for PVDF/PVP hollow fibers, the figures in Table 3 show an increasing retention with higher $M_{W}$
_PVP_. On the contrary, other researchers [64,65] report increasing retention and decreasing permeability for PVDF/PEG flat sheet and hollow fiber membanres by incrementing the PEG molecular weight. The presented results are in agreement with [64,65] and with the permeability/retention trade-off, thus, more research may be required into the influence of $M_{W}$
_PVP_ on the retention. As both *P* and *R* largely depend on the pore size, *c*_PVDF_ and $M_{W}$
_PVP_ seem to have an influence on the pore size. However, as neither the Hagen-Poiseuille relation for *P*, nor the Ferry-Rankin equation [1] for *R* are valid here—both theoretical models are only valid for long and cylindrical, not for asymmetric pores—no direct conclusions on the actual pore size may be drawn from the presented linear regression models for *P* and *R*.

Moreover, the non-significance of the remaining factors does not suggest that these do not have an influence on the membrane production process—their relevance has been demonstrated in literature [3,5,6]—but that these factors do not show an independent linear behavior in the studied range, i.e., there may be co-dependencies or non-linearities. For example, the PVP content may be favorable for the permeability if its molecule chains are short, i.e., $M_{W}$
_PVP_ is low, and it can be leached out from the polymer matrix during phase inversion, but may compromise permeability when the PVP chains are long and the pore interconnection pathways are increasingly blocked by embedded PVP molecules. In order to investigate such co-dependencies, a more detailed experimental design might lead to further insights. In terms of model fit, both models show a reasonable $R^{2}$, but a lower $Q^{2}$ with a difference of more than 0.3, which impairs the future prediction precision. Additionally, the model for *P* shows high reproducibility with low centre point variance, which may explain the low model validity (as pointed out in the discussion in Section 3.1.1). Similarly, the model validity is higher for the model for *R*, but the reproducibility is lower and the centre point divergence is higher. Overall, the models for *P* and *R* seem rather imperfect, but reasonable for being derived from a relatively inexpensive screening design.

The regression hyperplanes for *P* and *R* as functions of *c*_PVDF_ and $M_{W}$
_PVP_ are revealed in Figure 1 and Figure 2, respectively. High PVDF concentration and PVP molecular weight lead to low permeability and, vice versa, high macromolecule retention, which is displayed by the model and confirmed by the experiments (black dots). Where less than two dots are shown, the viscosity/solubility was the limiting factor for not carrying out the missing experiments. The regression hyperplanes in Figure 1 and Figure 2 agree with the permeability/retention trade-off. Moreover, it is noted that the chosen screening design is not suitable for parameter optimization. Hence, the presented models may allow to steer the membrane characteristics through variation of the PVDF content and the PVP molecular weight, but no parameter optimum can be derived from the linear regression models.

Besides, the descriptive statistics for the burst pressure are also presented in Table 2. The maximum burst pressure is 4.5 bar, whereas the majority of the tested membranes lies in the range of 2 bar to 3 bar, thus indicating the suitability for low-pressure micro-/ultrafiltration. However, no statistically significant linear model for the burst pressure could be found. Hollow fiber membranes made of similar PVDF concentrations and PVP co-polymer showed burst pressures of only 1.6 bar to 1.8 bar (pure water flux at 1 bar: approx. 160 L/$m^{2}$/h), highlighting the mechanical stability of the multi-channel design, although it is acknowledged that less material was used in the hollow fiber production [66].

#### 3.1.3. Scanning Electron Microscopy

The cross-sectional morphology of PVDF/PVP multi-channel membranes is shown in Figure 3. Firstly, Figure 3a reveals a regular and circular development of the feed channels with the outer and channel diameters in the range of the spinneret outlet dimensions (4.8
mm/0.9
mm) and finger-pores at the inner channel surfaces. Wan et al. [46], who studied seven-bore ultrafiltration membranes fabricated from the system PVDF/polyethylene glycol (PEG, $M_{W}$ = 3.35
kDa)/*N*-methyl-2-pyrrolidone (NMP) found lower cut-off values, but—in terms of morphology—a less uniform channel development, which strikes the importance of a customized spinneret design surpressing uncontrollable shear and promoting molecule chain alignment [2,63,67].

Furthermore, the close-ups in Figure 3b–d show representative examples of membranes with different PVP molecular weight. These images unfold the role of the PVP molecular weight in membrane formation. Whilst the morphology varies only slightly from $M_{W}$
_PVP_ = 10 kDa to $M_{W}$
_PVP_ = 55 kDa—solely the deeper void structures enlarge—all membranes fabricated with $M_{W}$
_PVP_ = 360 kDa exhibit finger-pores on the outer surface as well. As the nascent fibre emerges from the spinneret, phase inversion on the outer surface is induced through the applied steam in the gap between spinneret and coagulation bath [68]. However, if long PVP chains are present in the dope solution, increased shear forces at the spinneret outlet take place and, consequently, an enhanced die-swell effect ocurrs through polymer chain relaxation [69]. This may lead to ruptures in the formed skin layer, which propagate to finger-pores if the PVP molecular weight is too high.

#### 3.1.4. Atomic Force Microscopy

For AFM analysis, two membranes with a reasonable performance in terms of permeability and retention were selected (ID10: *P* = 132.1
L/$m^{2}$/h/bar and *R* = 54.5% and ID20: *P* = 187.8
L/$m^{2}$/h/bar and *R* = 42.2%). The correctness of the AFM images was confirmed by matching at least two subsequent surface scans in which the cantilever oscillation direction was set to 0 and 90, respectively, in order to preclude possible artefacts from AFM imaging, e.g., sample dragging. As a validation of the method, a multi-channel membrane from a previous work [49] prepared from PES/PVP/DMAc with a bore-fluid composition of 80% H_2_O:20% EtOH was also analyzed, yielding a root mean square (RMS) roughness of 10.57
nm (surface plot not shown), which compares to a value of 13.0
nm measured in the previous study.

The two-dimensional elevation plots of the central channel inner surface of the selected PVDF/MCM membranes are shown in Figure 4. In both samples, ridges and valleys parallel to the channel/spinning direction become apparent. An estimation of the Reynolds number Re in the bore-fluid flow during the phase inversion process yielded Re=251, i.e., laminar flow regime. Hence, any potential instability at the channel surface during the phase inversion process may be carried downstream by the bore-fluid and result in such valley/ridge structures. Besides, Khayet et al. [70] stated that the parallel structures at hollow fiber surfaces may be ascribed to high molecule chain orientation and alignment, which in this case, may be induced by the favorable spinneret design.

Moreover, the surface roughness is more than twice as high for ID10 compared to ID20. The reason for this may be multifaceted, as all process parameters except the PVDF molecular weight vary between the two samples, see Table 1. Singh et al. [71] reported that higher pore sizes lead to higher surface roughness. As particularly the PVP molecular weight plays a role in the morphology (see Figure 3), the difference in $M_{W}$
_PVP_ may explain the deviation of surface roughness as well. The values for $M_{W}$
_PVP_ were 10 kDa and 55 kDa for Figure 4a,b, respectively. Although the trend in *P* and *R* indicates less interconnected porous networks when the PVP molecular weight is increased, the actual pore size at the surface may be higher for longer PVP chains, as observed in Figure 4 and implied by Singh et al. [71]. Morevoer, the PVDF content was 18 wt.% and 15 wt.% for Figure 4a,b, respectively. As the PVDF content influences the overall porosity, this may be another reason for the difference in surface roughness. As the PVDF content influences the overall porosity, this may be another reason for the difference in surface roughness—a higher PVDF content provokes lower porosity and may consequently lead to lower surface roughness [72]. More research is required to scrutinize the interplay of PVP molecular weight, PVDF content, pore size, and surface roughness. In general, the higher surface roughness of the presented PVDF/PVP membranes compared to PES/PVP multi-channel membranes [49] may indicate a higher fouling susceptibility.

### 3.2. Variation of Bore-Fluid

Membrane characteristics are affected by the type and composition of the coagulation medium [5]. The influence of a varied bore-fluid composition, i.e., the inner coagulation medium, on the performance of ID10 (Table 1) is displayed in Figure 5. The retention decreases at organic compound (EtOH or DMAc) addition, whereas the permeability decreases for EtOH addition and increases slightly for DMAc addition. Deshmukh and Li [73] also observed a decrease in permeability with ethanol addition to the bore-fluid for PVDF/PVP gas permeation hollow fiber membranes and explained the observation with a decrease in porosity. They also found an increase in pore size, which may explain the decrease in retention here. In general, a denser skin layer is formed when a strong coagulant such as water is applied, whereas bigger pores, but lower porosity seem to be the result when weaker coagulants, such as ethanol or DMAc are used.

This argument is supported by SEM analysis of the membranes with varied organic content in the bore-fluid, shown in Figure 6. The secondary structure changes drastically, as finger-like voids (Figure 3b) are replaced with a denser and sponge-like structure. This may be retraced to a differences in the phase inversion process. The delayed phase inversion process, also referred to as solid-liquid demixing, brings about more sponge-like (Figure 6), but globule-type secondary microstructures, as opposed to instantaneous phase inversion (also liquid-liquid demixing), which causes finger-like macrovoids (Figure 3b) with cellular microstructure [51,74]. Even though the pore size at the active filtration surface, which determines the retention performance, could not be studied with the presented method, the pores at the inner channel surface seem to increase in size (*R* decreases). The combination of increased pore size at the inner surface, decreased porosity, and denser secondary structures lead to lower retention, but similar permeability, as shown in Figure 5.

### 3.3. Post-Treatment

Oxidative degradation and subsequent removal of the co-polymer PVP via sodium hypochlorite post-treatment (PT) is a well-known technique in membrane modification and the PVP degradation mechanism is elucidated in [75,76]. The effect of NaClO PT as a function of different PVP molecular weight has been investigated and is presented in Figure 7. The studied membrane samples were ID10, ID20, and ID6 (Table 1) with a PVP molecular weight of 10 kDa, 55 kDa, and 360 kDa, respectively. The permeability increases through NaClO PT, most notably for higher $M_{W}$
_PVP_. An increase in *P* and a decrease in *R* was also observed by Pellegrin et al. [77] for NaClO PT of PES hollow fiber membranes with PVP additive. A possible explanation for the $M_{W}$
_PVP_-dependence of the post-treatment effect is the amount of PVP that remains in the membrane matrix after phase inversion. Low $M_{W}$
_PVP_ mostly migrates to the coagulation medium during phase transition due to higher solubility in the non-solvent water, while high $M_{W}$
_PVP_ has lower non-solvent solubility, may be sterically hindered and remains entrapped in the PVDF network [50], as previously implied in the discussions of the effect of $M_{W}$
_PVP_ on *P*, *R*, and the morphology. Hence, the effect of degenerative PVP removal on the permeability would be much more pronounced for a high $M_{W}$
_PVP_, as obstructed pore interconnection paths may be opened, leaving behind a more porous network. Interestingly, the relative decrease in retention through NaClO PT in Figure 7 does not depend on the PVP molecular weight. This implies that the pore size increases through hypochlorite treatment (confirmed by [77]), regardless of the PVP molecular weight. Accordingly, hypochlorite PT leads to an increase in pore size—i.e., retention decreases—and, in the case of a high $M_{W}$
_PVP_, clears blocked pore interconnections—i.e., permeability increases.

The observed degenerative removal of PVP indicates that the hydrophilicity of the membrane may decrease if the membrane is operated in a long term or, particularly, if hypochlorite cleaning is applied to remove clogging and bacteria. The decreased hydrophilicity may, in turn, worsen the fouling behavior. Hence, more studies into the effect of NaClO on PVDF/PVP multi-channel membranes, particularly with regards to hydrophilicity and fouling, are recommended. Besides, it is noted that not only the PVP molecular weight is varied between the three samples, but also the PVP concentration, the PVDF concentration, and the process temperatures, see Table 1. This highlights an issue with the type of screening design applied in the present study—the study of individual parameter effects is more intricate than in full factorial designs.

## 4. Conclusions

Multi-channel membranes may provide enhanced productivity per volume and better mechanical stability compared to flat-sheet and single-channel arrangements if designed properly. Particular interest lies in the fabrication of PVDF multi-channel membranes, as they provide enhanced chemical stability compared to, for instance, PES. After a brief analysis of the literature dealing with multi-channel configuration, we concluded that, due to the complexity of the multi-channel formation process, it may be beneficial to apply DoE for the development of PVDF/PVP seven-channel membranes. We generated a three-level fractional factorial parameter screening design over several process parameters and, in a first step, derived a linear regression model for the dynamic viscosity of the PVDF/PVP/DMAc dope solution. Furthermore, we established process conditions for the manufacturing of PVDF/PVP seven-channel capillary membranes. The permeability and macromolecule retention form a linear relation with the variables PVDF content and PVP molecular weight, which is related to changes in porosity and pore network interconnection, respectively. SEM and AFM studies gave insights into the membrane morphology, but the elucidation of the interaction of pore size and surface roughness with PVDF content and PVP molecular weight requires further research. Although the feed capillaries promote strong shear forces at the channel surface, the fouling propensity may vary strongly due to the variation in surface roughness, demanding further research into the fouling aspect. Addition of weaker coagulants to the bore-fluid resulted in similar permeability but lower retention, which may be caused by an increase in pore size, but a decrease in porosity and denser secondary structures. Post-treatment with hypochlorite causes PVP chain scission and leaching, which can be used to increase the permeability while slightly compromising retention, but may deteriorate hydrophilicity and fouling behavior. The applied approach using DoE for parameter screening yielded satisfactory membrane performance and linear models, but may be complemented with more detailed experimental designs to study parameter interactions, non-linearities, and parameter optima. The membranes produced in this study showed characteristics suitable for MF/UF, but may also find application in MBR or membrane contactors.

## Figures and Tables

**Figure 1 polymers-11-00463-f001:**
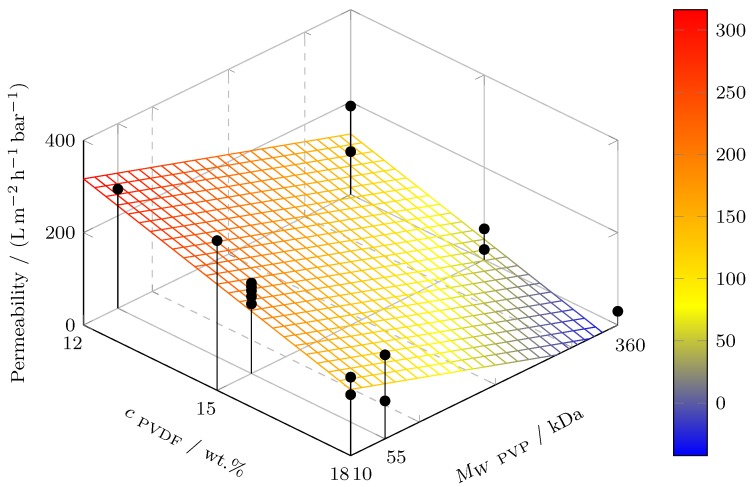
Regression hyperplane showing the linear fit of permeability to PVDF concentration and PVP molecular weight at *T*_Dope_ = 35 ${}^{\circ }C$, *T*_H_2_O_ = 35 ${}^{\circ }C$, *c*_PVP_ = 5%, $M_{W}$
_PVDF_ = 585 kDa. Black dots represent experimental values of entire design space (*n* = 16).

**Figure 2 polymers-11-00463-f002:**
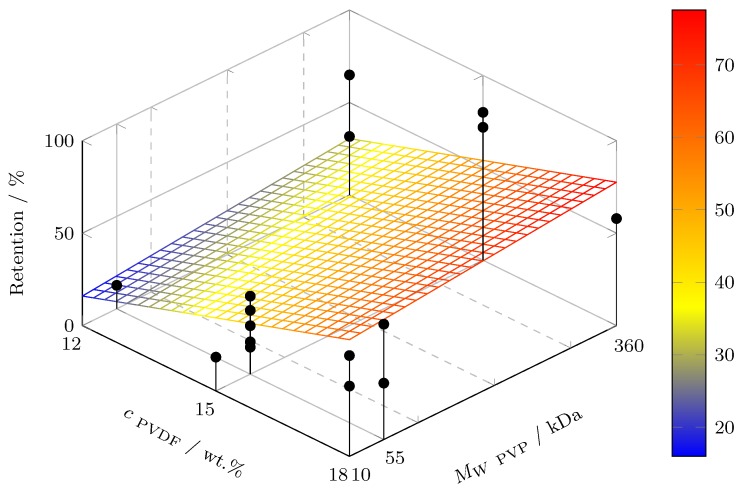
Regression hyperplane showing the linear fit of retention to PVDF concentration and PVP molecular weight at *T*_Dope_ = 35 ${}^{\circ }C$, *T*_H_2_O_ = 35 ${}^{\circ }C$, *c*_PVP_ = 5%, $M_{W}$
_PVDF_ = 585 kDa. Black dots represent experimental values of entire design space (*n* = 16).

**Figure 3 polymers-11-00463-f003:**
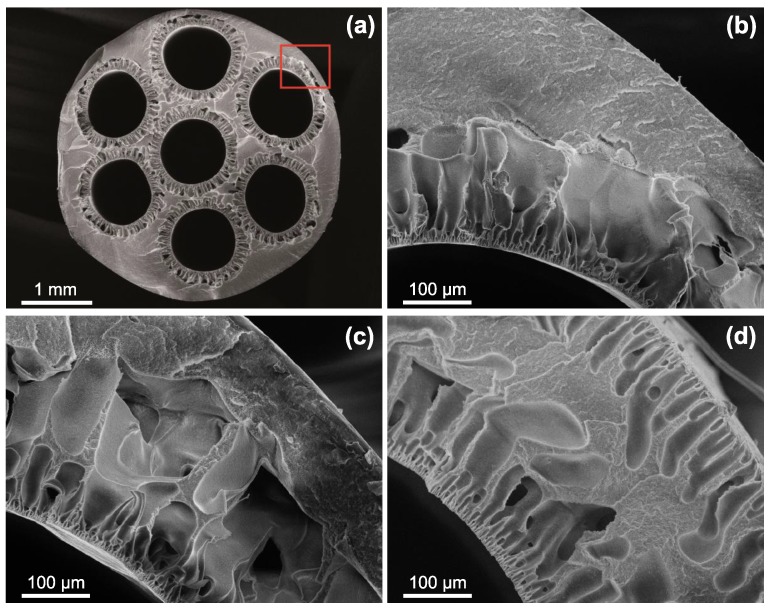
Cross sectional scanning electron microscope (SEM) images. (**a**,**b**) $M_{W}$
_PVP_ = 10 kDa (ID10); (**c**) $M_{W}$
_PVP_ = 55 kDa (ID5); (**d**) $M_{W}$
_PVP_ = 360 kDa (ID6). For detailed parameters see Table 1.

**Figure 4 polymers-11-00463-f004:**
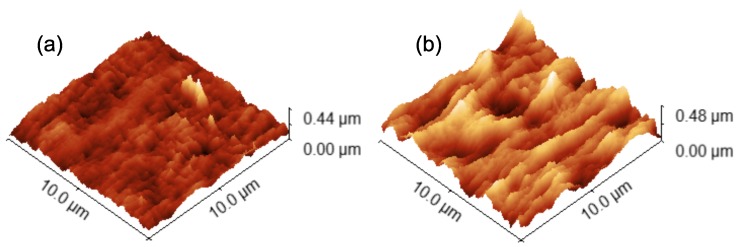
Atomic force microscopy (AFM) images of central channel surface. (**a**) Root mean square (RMS) roughness = 29.55
nm (grain masked for roughness calculation) (ID10); (**b**) RMS roughness = 68.58
nm (ID20). For detailed parameters see Table 1.

**Figure 5 polymers-11-00463-f005:**
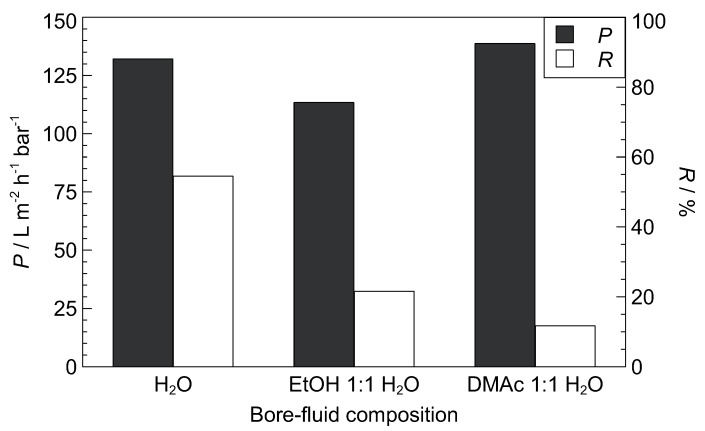
Permeability (*P*) and retention (*R*, dextran 500 kDa) of varied bore-fluid composition. H_2_O reference membrane is ID10 (Table 1).

**Figure 6 polymers-11-00463-f006:**
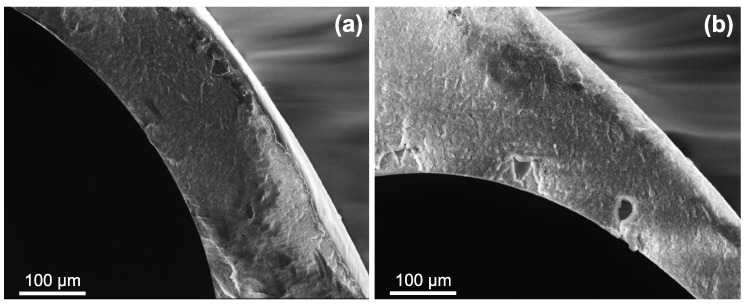
Cross sectional scanning electron mircoscope (SEM) images of varied bore-fluid composition. (**a**) ID10 with EtOH 1:1 H_2_O; (**b**) ID10 with DMAc 1:1 H_2_O. ID10 with pure H_2_O shown in Figure 3b. For detailed parameters see Table 1.

**Figure 7 polymers-11-00463-f007:**
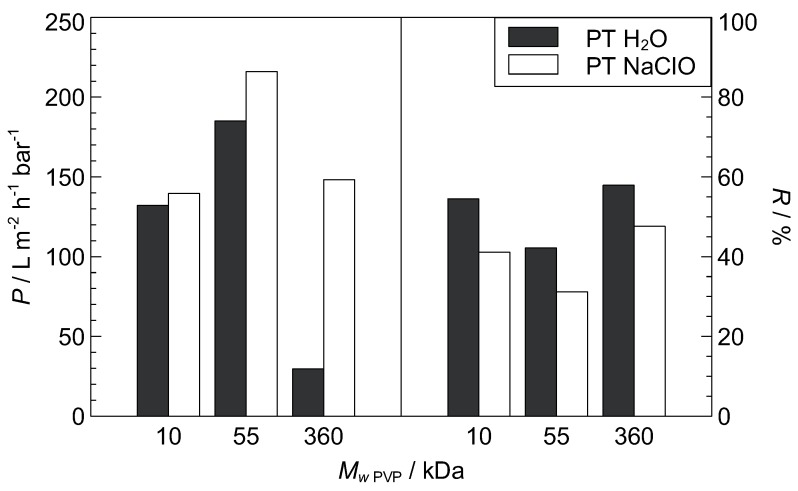
Influence of post-treatment (PT) conditions (1 day in H_2_O or 5000 ppm NaClO solution, respectively) on permeability (*P*) and retention (*R*, dextran 500 kDa) of membranes with varied PVP molecular weight

**Table 1 polymers-11-00463-t001:** Screening of dope solutions generated via Design of Experiments (DoE) and measured dyn. viscosity. Grey rows excluded from spinning experiments due to low viscosity.

ID	Run Order	*c*_PVDF_ wt.%	*c*_PVP_ wt.%	*T*_Dope_${}^{\circ }$C	*T*_H_2_O_${}^{\circ }$C	$M_{W}$_PVDF_ kDa	$M_{W}$_PVP_ kDa	η${}^{1}$ mPa s
1		12	4	20	20	390	10	819.9
2	5	15	4	20	35	585	55	4623
3		18	4	20	50	685	360	${}^{2}$
4		12	4	35	35	585	10	1229
5	12	15	4	35	50	685	55	6701
6	9	18	4	35	20	390	360	8353
7	6	15	4	50	20	685	10	4667
8	11	18	4	50	35	390	55	2393
9	14	12	4	50	50	585	360	2644
10	13	18	6	20	50	585	10	8634
11	3	12	6	20	20	685	55	4823
12	4	15	6	20	35	390	360	9256
13		15	6	35	50	390	10	1484
14	15	18	6	35	20	585	55	7627
15	10	12	6	35	35	685	360	11,220
16	16	18	6	50	35	685	10	10,540
17		12	6	50	50	390	55	550.4
18	7	15	6	50	20	585	360	7561
19	8	15	5	35	35	585	55	3326
20	2	15	5	35	35	585	55	3669
21	1	15	5	35	35	585	55	3211

${}^{1}$ measured at $\dot{\gamma }$ = 100 s^−1^; ${}^{2}$ no homogeneous solution.

**Table 2 polymers-11-00463-t002:** Basic statistics of investigated outcome variables.

	Dyn. Viscosity ^1,2^	Permeability ^3^	Retention ^3,4^	Burst Pressure ^3^
	mPa s	L m${}^{-2}$ h${}^{-1}$ bar ${}^{-1}$	%	bar
Min.	550	22.1	12.7	1.3
Max.	11,220	324.1	80.0	4.5
Mean	5167	152.0	41.1	2.5
Median	4645	168.5	36.3	2.1
Q1	2519	87.4	22.3	2.0
Q3	7990	189.6	60.2	3.1

${}^{1}$ measured at $\dot{\gamma }$ = 100 s^−1^; ${}^{2}$ Sample size n=20; ${}^{3}$ Sample size n=16; ${}^{4}$ of dextran 500 kDa.

**Table 3 polymers-11-00463-t003:** Multiple linear regression models.

	Dyn. Viscosity ${}^{1}$		Permeability		Retention ${}^{2}$	
	mPa s		L m${}^{-2}$ h${}^{-1}$ bar${}^{-1}$		%	
	**Coeff.**	p	**Coeff.**	p	**Coeff.**	p
Constant	−18,651.30	0.0000 *	787.43	0.0000 *	−136.69	0.0000 *
*c*_PVDF_/wt.%	812.56	0.0002 *	−28.85	0.0049 *	6.60	0.0141 *
*c*_PVP_/wt.%	959.67	0.0264 *	−12.62	0.3947	7.85	0.0775
*T*_Dope_/${}^{\circ }$C	−71.07	0.0451 *	1.91	0.1209	−0.32	0.3349
*T*_H_2_O_/${}^{\circ }$C			−0.48	0.6521	0.16	0.5873
$M_{w}$_PVDF_/kDa	14.26	0.0005 *	−0.18	0.3168	0.04	0.4105
$M_{w}$_PVP_/kDa	13.41	0.0001 *	−0.53	0.0009 *	0.13	0.0019 *
Model		0.0000 *		0.0035 *		0.0045 *
$R^{2}$	0.84		0.84		0.83	
$Q^{2}$	0.64		0.40		0.31	
Model validity	0.02		0.13		0.95	
Reproducibility	0.99		0.99		0.51	

${}^{1}$ measured at $\dot{\gamma }$ = 100 $s^{-1}$; ${}^{2}$ of dextran 500 kDa; * statistically significant (p<0.05).

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
