# Peer review of "Parameter Screening of PVDF/PVP Multi-Channel Capillary Membranes"

_polymers, 2019, doi:10.3390/polym11030463_

Round 1

Reviewer 1 Report

The manuscript provide very interesting results on PVDF/PVP Multi-Channel
Capillary Membranes. The experiments were designed and performed in a correct way by the authors; however, they should improve the material and method section and provide detailed information about the samples preparation for characterization. i.e. line 106 - for the SEM analysis the samples were covered with Au? line 109, which kind of tip was used for AFM investigation?

The manuscript is well-written, but the text should be re-read because there are several language mistakes, such as: 

 line 99 was used as post-treatment solution (missing "a" post-treatment solution)

line 203 instead of "A membrane’s permeability" should be "Membrane permeability"

or line 161 membrane’s characteristics...

Author Response

Response to reviewers – polymers-443472

To Note:  We have numbered the original reviewer comments (original black text below) and number linked these to the revised and marked version of the manuscript using the LaTeX package changes. Our responses below are in the blue text that follows the original reviewer comment. Any given line numbers refer to the revised and marked version of our manuscript. For display of the final LaTeX document please supply the option final to the package declaration, i.e. \usepackage[final]{changes} (line 71 of the .tex file), which clears the traces of changes and accepts our last changes.

Comments from Editor

The referees have noted a number of minor issues that should be addressed in a revision. In particular, additional discussion regarding the PVP component would be beneficial, as outlined by referee #2.

The comments made by the reviewers have been addressed point-by-point below. Particularly, further discussion regarding the co-polymer PVP has been added – please see comments 4-7 by Reviewer #2. Besides, no specific comments on the manuscript content were made by the Editor; we take this to be a positive outlook consistent with the ‘minor revisions’ requested.

Comments from Reviewers

Reviewer #1

The manuscript provides very interesting results on PVDF/PVP Multi-Channel Capillary Membranes. The experiments were designed and performed in a correct way by the authors; however, they should improve the material and method section and provide detailed information about the samples preparation for characterization.

1.     i.e. line 106 - for the SEM analysis the samples were covered with Au?

Thank you for the insightful comments. This comment was addressed by insertion of a phrase in line 110. For the used SEM system no coating with, for instance, Au is necessary and the samples can be analyzed directly.

2.     line 109, which kind of tip was used for AFM investigation?

Done – Information on the tip was added to the manuscript in line 116-118.

3.     The manuscript is well-written, but the text should be re-read because there are several language mistakes, such as: line 99 was used as post-treatment solution (missing "a" post-treatment solution) line 203 instead of "A membrane’s permeability" should be "Membrane permeability" or line 161 membrane’s characteristics...

The manuscript has been re-read carefully and minor language and spelling mistakes, such as the ones kindly mentioned, have been corrected.

Reviewer 2 Report

The concern is PVP is soluble in water, the produced fiber membrane will lose hydrophilicity or not if operated in a long term. Authors also used NaClO to remove pvp, which gives some benefit of the permeance. However, it might decrease the hydrophilicity of the membrane. Fouling might be easier to appear, how to handle this as a trade-off for permeance and hydrophilicity.

Is there any reaction between NaClO  and pvp during the posttreatment? Why did it make more significant Influence on the high molecular weight pvp derived membranes?

Cross-sections of Fig. 3 and Fig. 6 are totally different, what is the reason caused these different structures. It is better to include the water as bore fluid derived membrane in Fig. 6 as a comparison.

Author Response

Response to reviewers – polymers-443472

To Note:  We have numbered the original reviewer comments (original black text below) and number linked these to the revised and marked version of the manuscript using the LaTeX package changes. Our responses below are in the blue text that follows the original reviewer comment. Any given line numbers refer to the revised and marked version of our manuscript. For display of the final LaTeX document please supply the option final to the package declaration, i.e. \usepackage[final]{changes} (line 71 of the .tex file), which clears the traces of changes and accepts our last changes.

Comments from Editor

The referees have noted a number of minor issues that should be addressed in a revision. In particular, additional discussion regarding the PVP component would be beneficial, as outlined by referee #2.

The comments made by the reviewers have been addressed point-by-point below. Particularly, further discussion regarding the co-polymer PVP has been added – please see comments 4-7 by Reviewer #2. Besides, no specific comments on the manuscript content were made by the Editor; we take this to be a positive outlook consistent with the ‘minor revisions’ requested.

Reviewer #2

4.     The concern is PVP is soluble in water, the produced fiber membrane will lose hydrophilicity or not if operated in a long term.

We thank the reviewer for the constructive comments. The change in hydrophilicity has been addressed in point 5, see below. Moreover, it has been highlighted in the methodology that the membranes were washed several times with water (line 146) before testing. This would have removed any PVP that is not embedded in the membrane matrix, and changes in hydrophilicity during the experiments are deemed unlikely. 

5.     Authors also used NaClO to remove pvp, which gives some benefit of the permeance. However, it might decrease the hydrophilicity of the membrane. Fouling might be easier to appear, how to handle this as a trade-off for permeance and hydrophilicity.

This constructive comment has been incorporated into the discussion in Section 3.3, line 361-365, and into the conclusion (line 390). Although the study of the hydrophilicity was not within the scope of this work, it might have an important influence on the fouling behavior.

6.     Is there any reaction between NaClO  and pvp during the posttreatment?

The PVP degradation mechanism has been elucidated by Wienk et al. [75] (new reference added) and Pellegrin et al. [76]. Their studies have been referenced in line 343.

7.     Why did it make more significant Influence on the high molecular weight pvp derived membranes?

The mobility of PVP during the phase inversion process is thought to vary for different PVP molecular weights due to differences in solubility and the entrapment of longer PVP chains. Consequently, different amounts of PVP remain in the PVDF matrix after phase inversion. A subsequent NaClO treatment would then cause larger amounts of PVP to leach out from the PVDF matrix, leaving behind a more porous network. We carefully considered this comment and clarified our explanation in the text in section 3.3, line 350 and the following lines.

8.     Cross-sections of Fig. 3 and Fig. 6 are totally different, what is the reason caused these different structures. It is better to include the water as bore fluid derived membrane in Fig. 6 as a comparison.

The change in structure between Fig. 3 and Fig. 6 is caused by differences in the phase inversion process due to ethanol/DMAc addition to the bore-fluid. Weak solvent addition to the bore fluid causes transition from liquid-liquid demixing, which leads to finger-like pores, to solid-liquid demixing, which results in sponge-like structures. This has been reported by Sukitpaneenit and Chung [51,74] for PVDF hollow fibre membranes and observed in the presented study for PVDF multi-channel membranes. We have inserted some text around line 332 and now refer to Fig 3b in both the text and the caption of Fig 6 in order to provide better readability. We have, however, refrained from copying Fig. 3b to Fig. 6 and instead improved the caption of Fig. 6.

Reviewer 3 Report

The authors fabricated the seven-channel capillary membranes in a steam-dry-wet spinning process, while varying the composition of the polymer solution and the process temperatures in a three-level fractional factorial linear screening design. The polymers polyvinylidene flouride (PVDF) was the chemically resistant main polymer and polyvinylpyrrolidone (PVP) was added as hydrophilic co-polymer. The obtained membranes may be suitable for micro-/ultrafiltration and, at the same time, demonstrate the merits and limitations of DoE for multi-channel membrane screening. However, there are some points which still need to be clarified.

Regarding the membrane advantages, more references should be discussed in Introduction (see 10.1016/j.cherd.2017.11.024). What are the main merits compared to other methods?

 Line 44, 'GE Water' could be 'water'.

Line 51, please do not call 'Chung’s researchers'. Please correct the paragraph due to all the contents from the same group.

Fof DoE, I would suggest the authors to do a parametric optimization or the modeling. 

Please change 'wt.-%' to 'wt.%' or 'wt%'.

What is the reference for Eq. 1? 

In Table 2, what is the sample size unit?

Please double check the unit of the permeability in the whole text, L m2 h-1 bar-1 or m-2 h-1 bar-1?

In Line 342, why the results were different from previous results? Please add the reasons for the difference.

What are the best parameters of the membrane design?

Author Response

Response to reviewers – polymers-443472

To Note:  We have numbered the original reviewer comments (original black text below) and number linked these to the revised and marked version of the manuscript using the LaTeX package changes. Our responses below are in the blue text that follows the original reviewer comment. Any given line numbers refer to the revised and marked version of our manuscript. For display of the final LaTeX document please supply the option final to the package declaration, i.e. \usepackage[final]{changes} (line 71 of the .tex file), which clears the traces of changes and accepts our last changes.

Comments from Editor

The referees have noted a number of minor issues that should be addressed in a revision. In particular, additional discussion regarding the PVP component would be beneficial, as outlined by referee #2.

The comments made by the reviewers have been addressed point-by-point below. Particularly, further discussion regarding the co-polymer PVP has been added – please see comments 4-7 by Reviewer #2. Besides, no specific comments on the manuscript content were made by the Editor; we take this to be a positive outlook consistent with the ‘minor revisions’ requested.

Reviewer #3

The authors fabricated the seven-channel capillary membranes in a steam-dry-wet spinning process, while varying the composition of the polymer solution and the process temperatures in a three-level fractional factorial linear screening design. The polymers polyvinylidene flouride (PVDF) was the chemically resistant main polymer and polyvinylpyrrolidone (PVP) was added as hydrophilic co-polymer. The obtained membranes may be suitable for micro-/ultrafiltration and, at the same time, demonstrate the merits and limitations of DoE for multi-channel membrane screening. However, there are some points which still need to be clarified.

9.     Regarding the membrane advantages, more references should be discussed in Introduction (see 10.1016/j.cherd.2017.11.024). What are the main merits compared to other methods?

We thank the reviewer for the perceptive comments. The suggested reference has been added to the discussion of hollow fibre membranes in line 32. Moreover, we have inserted the merits and limitations of the chosen experimental design in line 83, which also relates to points 12 and 18. As for the rest of the introduction we have considered the comment carefully but have chosen not to implement any further changes related to this comment. The merits of the chosen membrane type is discussed thoroughly in the introduction, see the highlighted text in line 30 (benefits of a tubular geometry) and 37 (advantages of multi-channel designs). Moreover, we present the first overview of literature on multi-channel membranes in our introduction (line 36-74), resulting in a total number of 76 references, which we think is a relatively high number for a research paper.

10.     Line 44, 'GE Water' could be 'water'.

We opine that being the proper name of a company, ‘GE Water’ is capitalized. However, we did leave a comment for the copy editor to double-check this point. Furthermore, we have added that the company GE Water has been absorbed by ‘SUEZ Water Technology & Solutions’ and that Inge has been absorbed by BASF.

11.     Line 51, please do not call 'Chung’s researchers'. Please correct the paragraph due to all the contents from the same group.

Done – We modified the text in lines 52 and 64 so that this issue is addressed. However, we do believe that naming Prof. Chung is important in this section as it adds essential information and structure to the introduction.

12.     For DoE, I would suggest the authors to do a parametric optimization or the modeling.

In our discussion of comment 9 we added the merits and limitations of the chosen experimental design to the introduction in line 83, including the fact that the chosen screening design is not suitable for optimization. We added some text to discuss this point in detail in Section 3.1.2, line 253-257, and to the Conclusion (line 394). In short, besides the unsuitability of the screening design for parameter optimization, the permeability/retention trade-off makes a statement on the optimum parameter setting impossible.

13.     Please change 'wt.-%' to 'wt.%' or 'wt%'.

Done – has been changed to wt.% in both text and figures 1&2.

14.     What is the reference for Eq. 1?

Done – a reference for the linear regression model has been added.

15.     In Table 2, what is the sample size unit?

Table 2 has been edited through the insertion of table footnotes for the convenience of the reader and lack of clarity in unit definitions has been avoided.

16.     Please double check the unit of the permeability in the whole text, L m2 h-1 bar-1 or m-2 h-1 bar-1?

Done – the unit of the permeability is now L m-2 h-1 bar-1 in the entire text and in all tables and figures. The pure water flux in line 263 has the unit L m-2 h-1.

17.     In Line 342, why the results were different from previous results? Please add the reasons for the difference.

We have reconsidered our argument and found that Pellegrin et al. [77] studied the NaClO post-treatment of membranes with PVP, but did not vary the PVP molecular weight. They did find an increase in pore size with hypochlorite post-treatment, which agrees with our findings and is now clarified in the text around line 357.

18.  What are the best parameters of the membrane design?

See comment 12. We have been able to establish general process conditions for the manufacturing of PVDF/PVP seven-channel capillary membranes (see line 367), but parameter optimization was not within the scope of this study.

Round 2

Reviewer 2 Report

Authors have addressed all of my comments.